# Theoretical Study of Phase Behaviors of Symmetric Linear B_1_A_1_B_2_A_2_B_3_ Pentablock Copolymer

**DOI:** 10.3390/molecules28083536

**Published:** 2023-04-17

**Authors:** Bin Zhao, Qingshu Dong, Wei Yang, Yuci Xu

**Affiliations:** 1Department of Physics, Taizhou University, Taizhou 318000, China; 2State Key Laboratory of Molecular Engineering of Polymers, Department of Macromolecular Science, Fudan University, Shanghai 200433, China; 3Faculty of Materials Science and Chemical Engineering, Ningbo University, Ningbo 315211, China

**Keywords:** block copolymer, self-assembly, phase diagram, self-consistent field theory (SCFT)

## Abstract

The nanostructures that are self-assembled from block copolymer systems have attracted interest. Generally, it is believed that the dominating stable spherical phase is body-centered cubic (BCC) in linear AB-type block copolymer systems. The question of how to obtain spherical phases with other arrangements, such as the face-centered cubic (FCC) phase, has become a very interesting scientific problem. In this work, the phase behaviors of a symmetric linear B_1_A_1_B_2_A_2_B_3_ (*f*_A1_ = *f*_A2_, *f*_B1_ = *f*_B3_) pentablock copolymer are studied using the self-consistent field theory (SCFT), from which the influence of the relative length of the bridging B_2_-block on the formation of ordered nanostructures is revealed. By calculating the free energy of the candidate ordered phases, we determine that the stability regime of the BCC phase can be replaced by the FCC phase completely by tuning the length ratio of the middle bridging B_2_-block, demonstrating the key role of B_2_-block in stabilizing the spherical packing phase. More interestingly, the unusual phase transitions between the BCC and FCC spherical phases, i.e., BCC → FCC → BCC → FCC → BCC, are observed as the length of the bridging B_2_-block increases. Even though the topology of the phase diagrams is less affected, the phase windows of the several ordered nanostructures are dramatically changed. Specifically, the changing of the bridging B_2_-block can significantly adjust the asymmetrical phase regime of the *Fddd* network phase.

## 1. Introduction

The nanoscale nanostructures self-assembled from the block copolymer (BCP) systems have attracted widescale interest due to their potential technological applications in a variety of fields and in daily life [1,2,3,4,5]. For example, the self-assembled structures from BCPs can be used as organic electronics [6], for water purification [7], energy materials [8], photonic materials [9], as well as in drug delivery [10,11]. Generally, BCPs are composed of chemically distinct homopolymer connected by covalent bonds. The repulsive interaction between the different chemical blocks leads to phase separation; however, the connectivity of the blocks drives the phase to separate in the microscopic scale. From a thermodynamic point of view, the formation of different nanoscopic domains is governed by the delicate balance of interfacial energy and stretching energy. Specifically, the stable nanoscale structures that are self-assembled from AB-type BCPs are body-centered cubic (BCC), hexagonal cylinder (C_6_), bicontinuous phases-Gyroid (G) and *Fddd* (*O*^70^), and lamella (L), determined by a competition between the two opposite trends [12,13,14,15]. Beyond these conventional structures, some vagarious nanostructures have attracted more and more attention in both experiment and theory, including the helical structure [16,17,18,19], quasicrystalline phase [20,21,22], hierarchical structure [23,24,25,26], Frank–Kasper phases [27,28,29,30,31], and so on. For example, many well-known biological systems are presented in the form of helical structures, including DNA and RNA. However, the origin of the homochirality is rarely understood. Recent research has shown that the double helical structure can be formed from a number of typical BCP systems both in theory and experiment, indicating that the BCP system can provide a powerful platform with which to understand and reveal the mysterious origin of the life. It has become a crucial issue to reveal the relationship between the initial chain topologies of multiblock copolymers and the final thermodynamically stable self-assembled nanostructures. In general, the architectures can significantly change the phase behaviors of the AB-type BCPs. However, the theoretical study shows that the phase behaviors of symmetric triblocks (ABA) and alternating (ABAB…) linear BCPs are relatively unchanged [32]. Therefore, it is generally agreed that the AB linear multiblock copolymers have similar phase diagrams.

In 2018, Sakurai and coworkers synthesized a series of S_1_BS_2_ asymmetric triblock copolymers to investigate the relationship between chain architecture and ordered structures [33]. Encouragingly, their experimental results showed that, even though the total molecular weight, volume fraction, and thermal annealing temperature were kept unchanged, the thermodynamically stable ordered structure was found to change by tuning the relative length between S_1_ and S_2_ blocks. These experimental results are consistent with Matsen’s prediction qualitatively, which is implemented by the self-consistent field theory (SCFT) [34]. Notice that the linear AB-multiblock copolymers have more tunable parameters to adjust the length ratio of each A/B-block; therefore, it provides a broad parameter space to regulate and research the self-assembly of AB-type BCP systems. Furtherly, Kim et al. investigated the phase behaviors of various linear tetrablock copolymers (S_1_I_1_S_2_I_2_) with fixed symmetric volume fraction (*f*_PS_ ≈ *f*_PI_ ≈ 0.5 and *f*_PS1_ ≈ *f* _PS2_) [35]. Interestingly, they observed the C_6_ and G phases by tuning the relative length ratio *f*_PI1_, which is also in accordance with the prediction of SCFT [36].

As one of the most successful methods, SCFT plays a critically important role to explain and predict the thermodynamically stable state of the BCP systems [37,38,39,40]. Due to its strong ability to calculate the free energy accurately as well as to visualize the distribution of each segment in the ordered phase, SCFT has been regarded as the standard theoretical tool for the study of the self-assembly of the BCPs. In 2014, Xie et al. predicted a series of novel binary mesocrystals from B_1_AB_2_CB_3_ multiblock terpolymers, including NaCl, CsCl, ZnS, α-BN, AlB_2_, CaF_2_, TiO_2_, and so on, in which a minority of A- and C-blocks self-assembled into the spherical domains and a majority of B-blocks formed the matrix [41]. The main idea is that the length of middle B_2_-block can control the average coordination number (CN) of the binary mesocrystals, while the asymmetry of CN is modulated by the relative length of the terminal B_1_- and B_3_-block. This distinguished work opens a window for the inverse design of BCPs to obtain the targeted structure, and thus makes it possible to obtain total binary crystals and even ternary crystals in the BCP systems. The reversed design principle was not only applied to ABC-type BCP systems, but also to the AB multi-BCP systems in recent years. 

Recently, several mechanisms of the self-assembly of BCPs has been proposed, including local segregation [30,42], the solubilization effect [34], packing frustration [43], a stretched bridge [41], and combinatorial entropy [44]. For example, our previous work predicted that the Frank–Kasper (FK) σ and A15 phases can be stabilized from a linear A_1_B_1_A_2_B_2_ tetrablock copolymer, in which the longer A_1_-block forms the “core” while the short A_2_-block forms the “shell” to regulate the sizes and shapes of the spherical domains, respectively [42]. Further, a linear B_1_A_1_B_2_A_2_B_3_ pentablock copolymer has been proposed to predict the single gyroid (SG) structure, in which a minority of A-blocks self-assemble into the network [45]. By tuning the asymmetry between the B_1_- and B_3_-blocks as well as the middle bridging B_2_-block, the phase transition from DG to SG can be triggered, utilizing the synergistic effect of two different mechanisms, i.e., packing frustration and stretched bridging effects. It is worth noting that the hexagonal cylinders (C_6_) are completely replaced by the unusual square array of cylinders (C_4_) as well as the rectangular array of cylinders (C_rec_). Because the relative lengths of the multiple blocks can be adjusted, this theoretical study demonstrates that the AB linear multi-BCPs can enrich the phase behaviors compared with simple AB and ABA copolymers. Encouragingly, some of the theoretical predictions were verified by the experiments. For example, the complex FK spherical phases have been predicted from conformational asymmetry AB-type BCP melts, in which the FK spherical domains have different sizes and shapes [29]. However, the theoretical prediction was not verified by experiments until 2017. In this experiment, Bates and co-workers tactfully used a series of di-BCPs (i.e., PEP-*b*-PLA, PI-*b*-PLA, and PEE-*b*-PLA) with a different degree of conformational asymmetry, verifying that the FK phases can only be formed under the condition of a highly asymmetric structure in melt systems [28]. With the close cooperation between theory and experiment, the explanatory ability and predictive power of SCFT have been further confirmed. 

However, up to now, the single stretched bridging effect on the AB linear multiblock copolymer self-assembly has not been considered. Recently, several typical (B_1_A_1_B_2_)*_m_* (*m* = 3, 5) star copolymers have been well studied systematically, showing that some intriguing phase behaviors have been predicted [46,47]. Particularly, the star architecture (B_1_A_1_B_2_)*_m_* will be converted to a symmetric linear B_1_A_1_B_2_A_2_B_3_ multi-BCP for *m* = 2. Due to a rare report about the stretching bridge in the linear AB multi-BCPs, in this paper, we focused on the symmetric linear B_1_A_1_B_2_A_2_B_3_ penta-BCP with fixed *f*_B1_ = *f*_B3_ and *f*_A1_ = *f*_A2_ to reveal the single stretched bridging effect on the phase behaviors. 

It is generally believed that the self-assembly of symmetrical AB-linear multi-BCPs usually forms a symmetrical phase diagram, and that the dominant spherical phase is BCC rather than other spherical phases. However, our SCFT calculations demonstrate that the length ratio of middle bridging B_2_-block plays a key role in modifying the self-assembly behaviors of the symmetrical B_1_A_1_B_2_A_2_B_3_ multi-BCP. On the one hand, the BCC can be replaced by FCC completely as the length ratio of the middle bridging B_2_-block is within a specific value range. On the other hand, the symmetry of the phase diagram is broken by changing the relative length of the B_2_-block. 

The following sections are arranged as follows. In Section 3, a detailed introduction of SCFT is presented. In Section 2, the results and a discussion are presented. First, the segment distribution of the relative length of the B_2_-block in lamellar phase is revealed. Then, we construct a series of phase diagrams in *f*_A_–*χN* plane with fixed *f*_A1_ = *f*_A2_, *f*_B1_ = *f*_B3_ and several typical *τ* = 1/2, 2/3, 4/5 (*τ* = (*f*_B1_ + *f*_B3_)/*f*_B_) to reveal the effects of the *f*_A_ and *χN* on the phase behaviors of the symmetric B_1_A_1_B_2_A_2_B_3_ pentablock copolymers. Further, we also prove that the relative ratio of B_2_-block can influence the phase behaviors with fixed *χN* = 100. Finally, we conclude this article in Section 4.

## 2. Result and Discussion

To make our results as reliable as possible, nine ordered structures in the multi-AB linear BCPs are considered in Figure 1. It should be noted that the hexagonal close packing sphere (HCP) is not considered in this paper, as it usually has an insignificant free energy difference compared with FCC and is located around the phase boundary [48].

### 2.1. Effect of Length Ratio of B_2_-Block on the Formation of L Structure

To the best of our knowledge, the A/B segment distributions have a marked effect on the formation of the ordered nanostructures [32]. First, we depict the possible segment distributions of the A/B block in the lamellar phases in Figure 2 to uncover the effect of length ratio of the bridging B_2_-block on the stability of the ordered lamellar nanostructures. Apparently, the A_1_- and A_2_- as well as B_1_- and B_3_-blocks have the same segment distributions, respectively, because of the symmetric architecture of the linear B_1_A_1_B_2_A_2_B_3_ penta-BCP with fixed *f*_A1_ = *f*_A2_ and *f*_B1_ = *f*_B3_. In general, the B_1_- and B_3_-blocks prefer to aggregate in the same domain with B_2_-block to form the B-domain in the lamellar phase shown in Figure 2b to reduce the loss of interfacial energy. Specifically, the B_1_- and B_3_-blocks aggregate into the different domain compared with B_2_-block as the length ratio of B_2_ block (1 − *τ*) is too long or too short, as shown in Figure 2a,c. In this case, a short B_1_/B_3_ (or B_2_) block will prefer to mix in the A-domain instead of the B-domain, leading to an extra interfacial energy cost but compensated with more entropy energy.

In Figure 3, the 1D segment distributions of the A/B block in lamellar phases are presented with three typical values *τ* = 0.1, 0.5, and 0.9, respectively, which are highly consistent with Figure 2. It is evidenced that the local phase segregation emerged regardless of whether the middle B_2_-block is too short or too long, resulting in the expanded domain spacing of the ordered nanostructures. Therefore, we can speculate regarding the segment distribution of A/B block in lamellar phases from the corresponding domain spacing shown in Figure 4. This means that the short B_1_/B_3_ and B_2_ block will aggregate into the A-domain for *τ* < 0.2 and *τ* > 0.8, respectively, while the B_1_/B_3_ and B_2_ blocks will aggregate into the same B-domain as the value of *τ* ranges from 0.2 to 0.8. This local segregation behavior has also been reported from asymmetric ABA and ABAB linear BCP systems [24,36]. Both experimental and theoretical research have confirmed that even a subtle change in the length ratio can strongly influence the phase transition.

### 2.2. Phase Diagram in f_A_-χN Plane with Three Typical τ

To reveal the effect of the middle bridging B_2_-block on the phase transition sequences, five typical values of *τ* (0, 0.1, 0.5, 0.8, 1) have been calculated for *χN* = 100 and *f*_A_ < 0.5, as shown in Figure 5. Interestingly, the phase transition point between G and L changes from 0.314, 0.223, 0.356, 0.290, to 0.350 as the *τ* increases. These devious phase transition points can be attributed to the changing of the effective volume fraction of the A/B component relative to their characteristic values. Specifically, the short B_2_ (or B_1_/B_3_) blocks will prefer to mix in the A-domain for *τ* = 0.8 (*τ* = 0.1), resulting in an enlarged effective volume fraction of the A-component; thus, the phase boundary between G and L shifts to the left side, as shown in Figure 5b,d. For the same reason, the phase boundary between G and C_6_ phases have similar trends. What is more noteworthy is that the BCC phase is replaced by FCC for *τ* =0.8 in Figure 5b.

Without the loss of the generality, three phase diagrams in the *f*_A_–*χN* plane with three typical *τ*_A_ = 1/2, 2/3, and 4/5 are constructed in Figure 6. Generally, the main features of these phase diagrams are similar to AB di-BCP as well as to ABA tri-BCP [32]. For *τ* = 1/2 and 2/3, the two phase diagrams are almost symmetrical, resulting from the symmetrical architecture of the AB linear multi-BCP and the uniform stretching of A- and B-blocks. As the *τ* increased to 4/5, the phase diagram becomes more asymmetric compared with the case of *τ* = 1/2 and 2/3, where the cylindrical and spherical phase regions expanded remarkably in the right part. This particular phase transition sequence is evidenced by the free energy comparisons of the candidate phases shown in Figure 7, indicating the phase sequence of L → G → C_6_ → FCC.

To uncover the underlying sophisticated phase transition mechanism between these spherical and cylindrical phases, not only the free energy, but also the corresponding interfacial energy and entropy energy comparisons, are plotted as a function of *f*_A_ in Figure 8a–c, respectively. This shows that the stable C_6_ phase region is replaced by spherical phase for *f*_A_ > 0.76, as the obtained interface energy from C_6_ phase cannot compensate the huge entropy loss. Figure 8a indicates that there is a very small free energy difference between the A15 and s phases, and the free energies of BCC and FCC phases are almost degenerative. The A15 and s phases have a favorable entropy energy while the loss of interfacial energy is larger than the BCC and FCC phases shown in Figure 8b,c, suggesting that the A15 and s phases are metastable in this symmetrical linear B_1_A_1_B_2_A_2_B_3_ penta-BCP system.

### 2.3. Phase Diagram in f_A_-τ Plane with Fixed cN

Generally, the phase diagrams in the *f*_A_–*τ* plane with fixed *χN* = 100 is presented in Figure 9. From the perspective of the changing tendency of these different phase regions, there is a significant difference between the two-dimensional (2D) and three-dimensional (3D) structures. This implies that the phase region of the 2D cylindrical phase changes in the range of 0.028 < Δ*f*_C_ < 0.179 is significant, while the changing of the phase regions of 3D gyroid (0.014 < Δ*f*_G_ < 0.052) and spherical (0.006 < Δ*f*_S_ < 0.073) phases are comparatively milder. Generally, the changing of the length ratio of the bridging B_2_-block will result in the non-uniform stretching of each B block, leading to the unfavorable entropy cost. Based on our understanding of self-assembly, the unfavorable stretching of each block is relieved more easily in 3D space compared with 2D space.

What is more noteworthy is the appearance of the phase transition sequence, i.e., BCC -> FCC-> BCC -> FCC-> BCC, shown in Figure 8 for fixed *χN* = 100. In order to uncover the phase transition mechanism between FCC and BCC structures, the detailed difference of free energy as well as entropy and interfacial energies are calculated and presented in Figure 10a,b, respectively. Generally, the FCC is less stable than the BCC, as the non-uniform stretching of the B-block leads to an extra packing frustration in the FCC. The FCC becomes stable over the BCC only when the gain of interfacial energy can compensate for the loss of entropy energy.

To further visualize and explore the stabilization mechanism of the FCC and BCC spherical phases, the isosurface plots of different A- and B-blocks in BCC and FCC spherical phases with six typical parameters are presented in Figure 11. The detailed parameters are listed below each subfigure. As the spherical domains can be formed from A- and B-components, respectively, these typical parameters can be divided into two categories (i.e., (a1) (b1) (c1) and (a2) (b2) (c2)). The first part is shown in the left panel, where the short A-blocks form spheres while the long B-blocks form the matrix. When *τ* ≈ 0.8 (Figure 11(a1)), the short B_2_-block aggregate near the A/B interface and the free end of the long B_1_- (and B_3_-) block can extend to the vertices of the Voronoi cell. The length difference between B_2_ and B_1_/B_3_ can reduce the packing frustration of B-blocks, thus stabilizing FCC. Similarly, when *τ* ≈ 0.3 (Figure 11(c1)), the short B_1_ /B_3_-blocks aggregate near the A/B interface, and the long B_2_-block extends to the vertices of the Voronoi cell. However, when *τ* ≈ 0.5 (Figure 11(b1)), *f*_B2_/2 ≈ *f*_B1_; therefore, none of the B blocks have the advantage of extending to the vertices of the Voronoi cell. As a result, the BCC is more stable than the FCC in this case. The second part is shown in the right panel, where the short (or part of) B- blocks form spheres and the long A- blocks form the matrix. When *τ* ≈ 0.8 (Figure 11(a2)), the short B_2_ block cannot be phase separated and dissolved in the A-matrix. However, as shown in Figure 11(a2), the concentration of B_1_ block increases at some special positions, thus reducing the interfacial energy. In the BCC phase, the B_1_-block prefers to aggregate at the edges of the Voronoi cell. In FCC phase, the B_1_-block prefers to aggregate at the vertices of the Voronoi cell, which is more concentrated than in the BCC. Therefore, the FCC phase is more stable than the BCC phase. The same is true in the case of *τ* ≈ 0.3 (Figure 11(c2)), since B_1_ and B_3_ blocks prefer to aggregate at the vertices of the Voronoi cell in FCC. When *τ* ≈ 0.5 (Figure 11(b2)), all B-blocks are long enough for phase separation from the A-domain. Therefore, the BCC phase is more stable than the FCC phase.

## 3. Theory and Method

In this study, the system consists of *n* B_1_A_1_B_2_A_2_B_3_ penta-BCP chains in the volume of *V*. Each penta-BCP consists of *N* segments with an equal segment length *b* and density *ρ*. The length of each block is specified as *f*_B1_*N*, *f*_A1_*N*, *f*_B2_*N*, *f*_A2_*N*, and *f*_B3_*N* (*τ* = (*f*_B1_ + *f*_B3_)/*f*_B_), respectively. The penta-BCP can be reduced to BAB or ABA, as *τ* equals 1 or 0. The variables *ϕ*_A_(r) and *ϕ*_B_(r) are used to characterize the distribution of the volume fraction of A-and B-blocks in the self-assembled mesoscopic structures. Under the approximation of the mean-field treatment and Gaussian chain model, the free energy per chain is in the unit of thermal energy *k*_B_*T* at the temperature *T*, and the *k*_B_ is the Boltzmann constant, which can be expressed as: (1)$$F/{\mathrm{nk}}_{B}T=-\mathrm{In}Q+\frac{1}{V}\int \mathrm{dr}\{\chi N\phi_{A}(r)\phi_{B}(r)-\omega_{A}(r)\phi_{A}(r)-\omega_{B}(r)\phi_{B}(r)-\eta (r)(1-\phi_{A}(r)-\phi_{B}(r))\}$$
where the *ϕ*_k_(r) is the mean field conjugate to *ϕ*_k_(r) (k = A or B). Moreover, *η*(r) is the Lagrange multiplier used to force the incompressibility condition: $\phi_{A}(r)+\phi_{B}(r)=1.0$. Moreover, the variable quantity *Q* is the partition function interacting with the mean field *ω*_A_ and *ω*_B_ produced by the surrounding chains, which is determined by
(2)$$Q=\frac{1}{V}\int \mathrm{dr}\;q(r,s)q^{+}(r,s)$$
where the *q*(r, s) and *q^+^*(r, s) are the propagator starting from the distinguishable ends, satisfying the following modifying diffusion equations:(3)$$\frac{\partial q(r,s)}{\partial s}=\nabla^{2}q(r,s)-\omega (r,s)q(r,s)$$
(4)$$-\frac{\partial q^{+}(r,s)}{\partial s}=\nabla^{2}q^{+}(r,s)-\omega (r,s)q^{+}(r,s)$$
where the *w*(r,s) = *ω*_k_(r), where s belongs to K blocks (K = A, B). In the above equations, the $R_{g}$ = (N/6)^1/2^*b* is chosen as the unit length. The initial conditions of the propagator functions are *q*(r,0) = 1.0 and *q^+^*(r,1) = 1.0. 

Minimization of the free energy leads to the following SCFT equation:(5)$$\omega_{A}(r)=\chi N\phi_{B}(r)+\eta$$
(6)$$\omega_{B}(r)=\chi N\phi_{A}(r)+\eta$$
(7)$$\phi_{B}(r)=\frac{1}{Q}[\int_{0}^{f_{B1}}\mathrm{dsq}(r,s)q^{+}(r,s)+\int_{f_{B1}+f_{A1}}^{f_{B1}+f_{A1}+f_{B2}}\mathrm{dsq}(r,s)q^{+}(r,s)+\int_{1-f_{B3}}^{1}\mathrm{dsq}(r,s)q^{+}(r,s)]$$
(8)$$\phi_{A}(r)=\frac{1}{Q}[\int_{f_{{}_{B1}}}^{f_{B1}+f_{A1}}\mathrm{dsq}(r,s)q^{+}(r,s)+\int_{f_{B1}+f_{A1}+f_{B2}}^{f_{B1}+f_{A1}+f_{B1}+f_{A2}}\mathrm{dsq}(r,s)q^{+}(r,s)]$$

The above equations are solved using the pseudo-spectral method, and the Anderson-mixing is implemented to accelerate the calculation [48,49,50,51,52,53]. The rectangular cubic is chosen to solve the periodic boundary conditions, where the lattice length is less than 0.1 *R*_g_ and the chain contour is divided to 200 points.

## 4. Conclusions

In summary, the self-assembly behaviors of a symmetric linear B_1_A_1_B_2_A_2_B_3_ penta-BCP have been studied using the SCFT. Several interesting self-assembly behaviors have been observed in the phase diagrams with respect to the *f*_A_–*χN* and *f*_A_–*τ* planes. One of the most striking features is the emergency of alternating phase transition sequences between the FCC and BCC phases as *τ* increases, indicating that the length of the B_2_-bridging block can play a sophisticated role to regulate the phase behaviors. It has been demonstrated that interfacial energy is a dominant factor in the stabilization of the FCC over the BCC. Particularly, the 3D complex FK σ and A15 spherical phases are less stable than the BCC and FCC phases, as the gain of entropy energy cannot compensate for the loss of interfacial energy in this symmetric B_1_A_1_B_2_A_2_B_3_ penta-BCP. Additionally, the phase region of the *O*^70^ phase is extremely influenced due to the non-uniform stretching of the bridging B_2_-blcok. Our SCFT calculations reveal the importance of the middle bridging B_2_-block in regulating the phase behaviors of symmetric linear B_1_A_1_B_2_A_2_B_3_ penta-BCP.

## Figures and Tables

**Figure 1 molecules-28-03536-f001:**
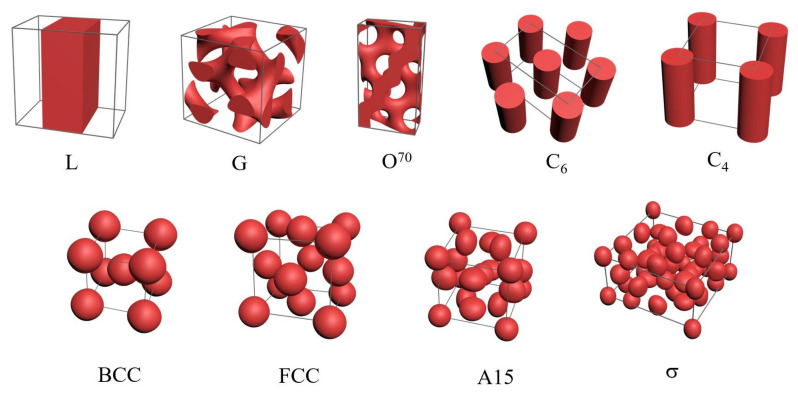
Schematic of candidate ordered phases considered in this work, including the lamellar (L) phase and networks of gyroid (G) and *Fddd* (*O*^70^) phases, two different cylinder phases (C_6_, C_4_), and four different spherical phases (BCC, FCC, A15, σ).

**Figure 2 molecules-28-03536-f002:**
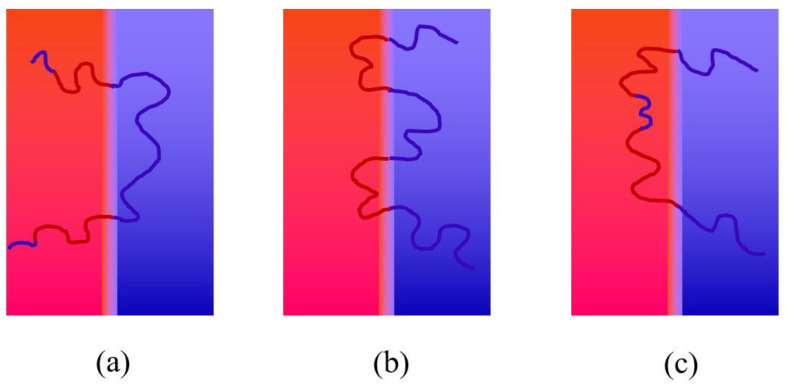
The plots of possible distributions of A- and B-blocks in lamellar phase, A and B blocks are shown in Red and Blue color respectively. (**a**) The short B_1_-(B_3_) and long B_2_-blocks segregate into the different domains; (**b**) The B_1_-(B_3_) and B_2_-blocks segregate within the same domains; (**c**) The long B_1_-(B_3_) and short B_2_-blocks segregate into the different domains.

**Figure 3 molecules-28-03536-f003:**
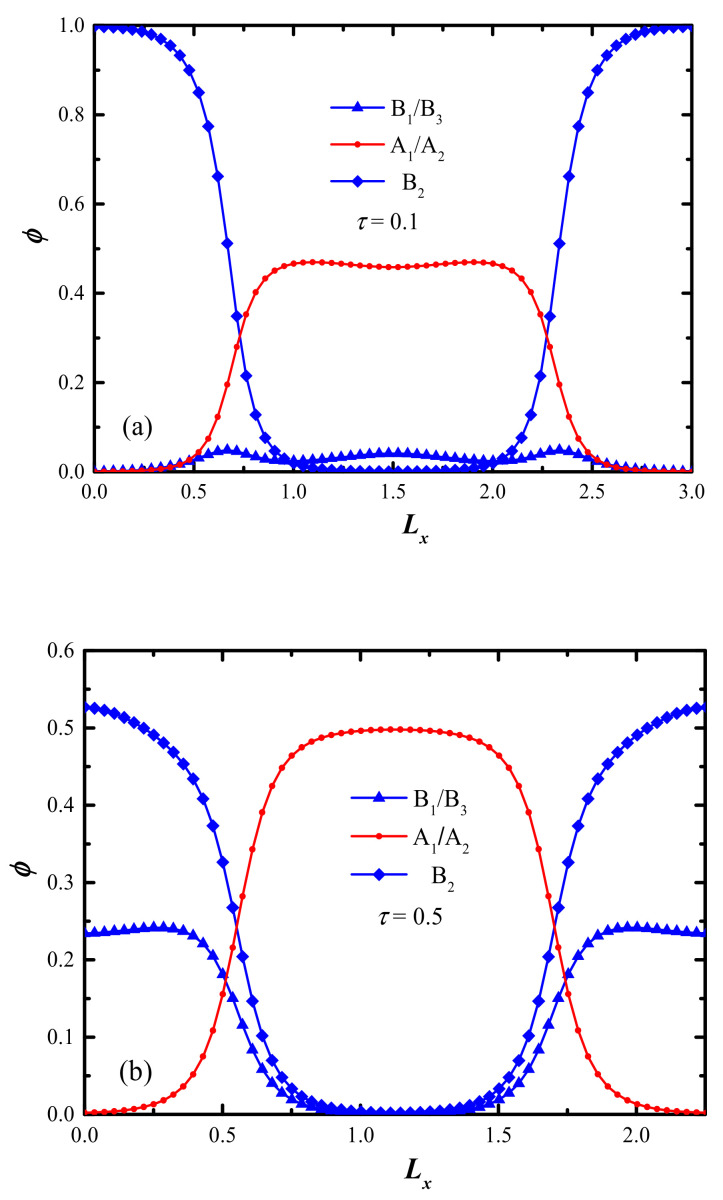
The comparisons of density profiles in the lamellar phase for three typical *τ* = 0.1, 0.5, and 0.9 shown in (**a**–**c**), respectively, where the B_1_/B_3_ block, A_1_/A_2_ block, and B_2_ block are shown in blue solid line with triangle symbol, red solid line with circle symbol, and blue solid line with square symbol, respectively.

**Figure 4 molecules-28-03536-f004:**
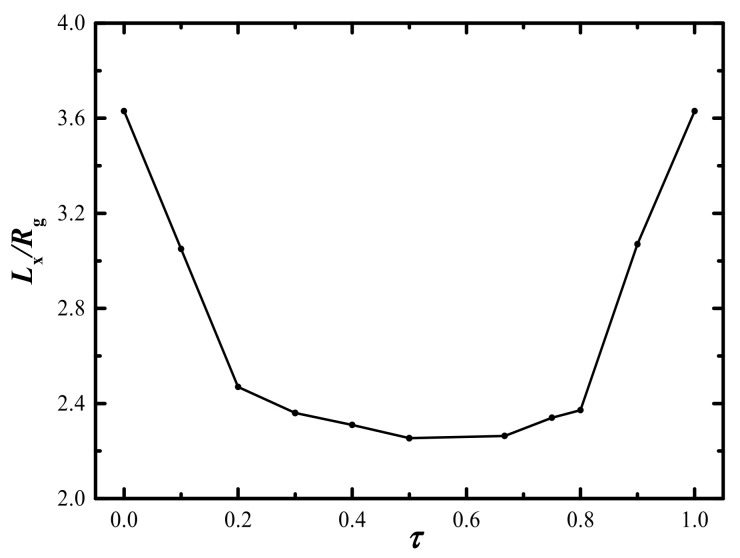
The period of lamellar phase as a function of *τ* for fixed *χN* = 100 and *f*_A_ = 0.5.

**Figure 5 molecules-28-03536-f005:**
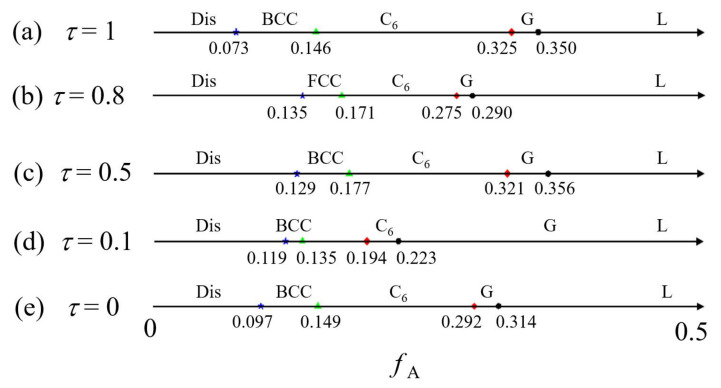
The phase transition sequences of the ordered nanostructures with fixed *χN* = 100 and *f*_A_ < 0.5 for several typical *τ* = 0, 0.1, 0.5, 0.8, and 1, respectively. The blue symbol indicate the phase transiton point between disorder and spherical phases, green symbol indicate the phase transiton point between spherical and cylindrical phases, red symbol indicate the phase transiton point between cylindrical and gyroid phases, and black symbol indicate the phase transiton point between gyroid and lamellar phases.

**Figure 6 molecules-28-03536-f006:**
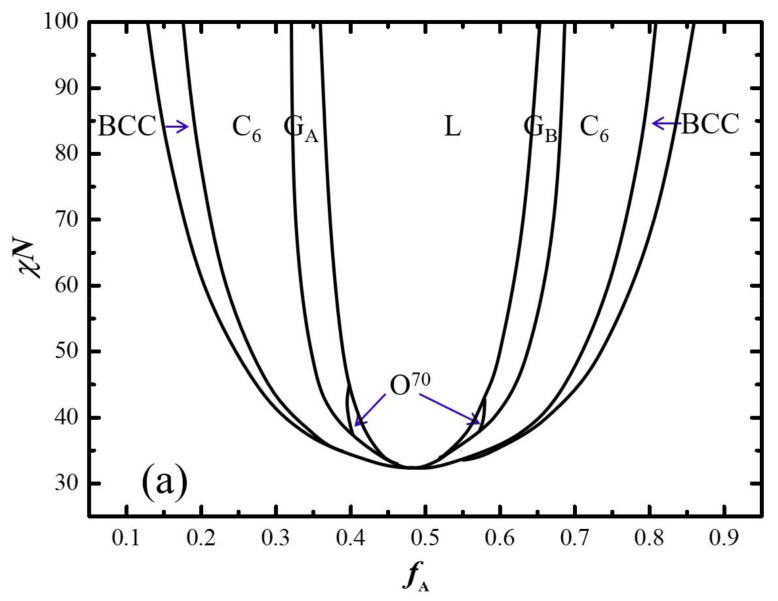
Phase diagrams in *f*_A_–*χN* plane with three typical *τ* = 1/2, 2/3, and 4/5 (**a**–**c**), respectively.

**Figure 7 molecules-28-03536-f007:**
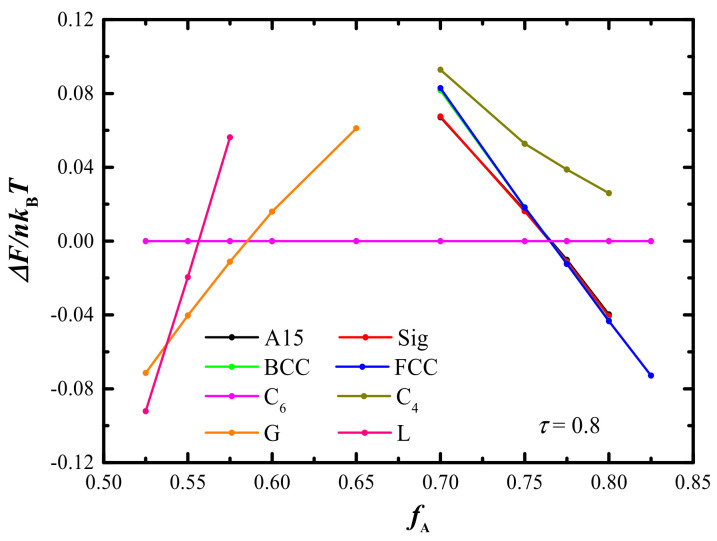
Comparisons of free energy along changing *f*_A_ of the candidate phases with fixed *τ* = 0.8 and *χN* = 80, where the free energy of C_6_ is treated as the reference.

**Figure 8 molecules-28-03536-f008:**
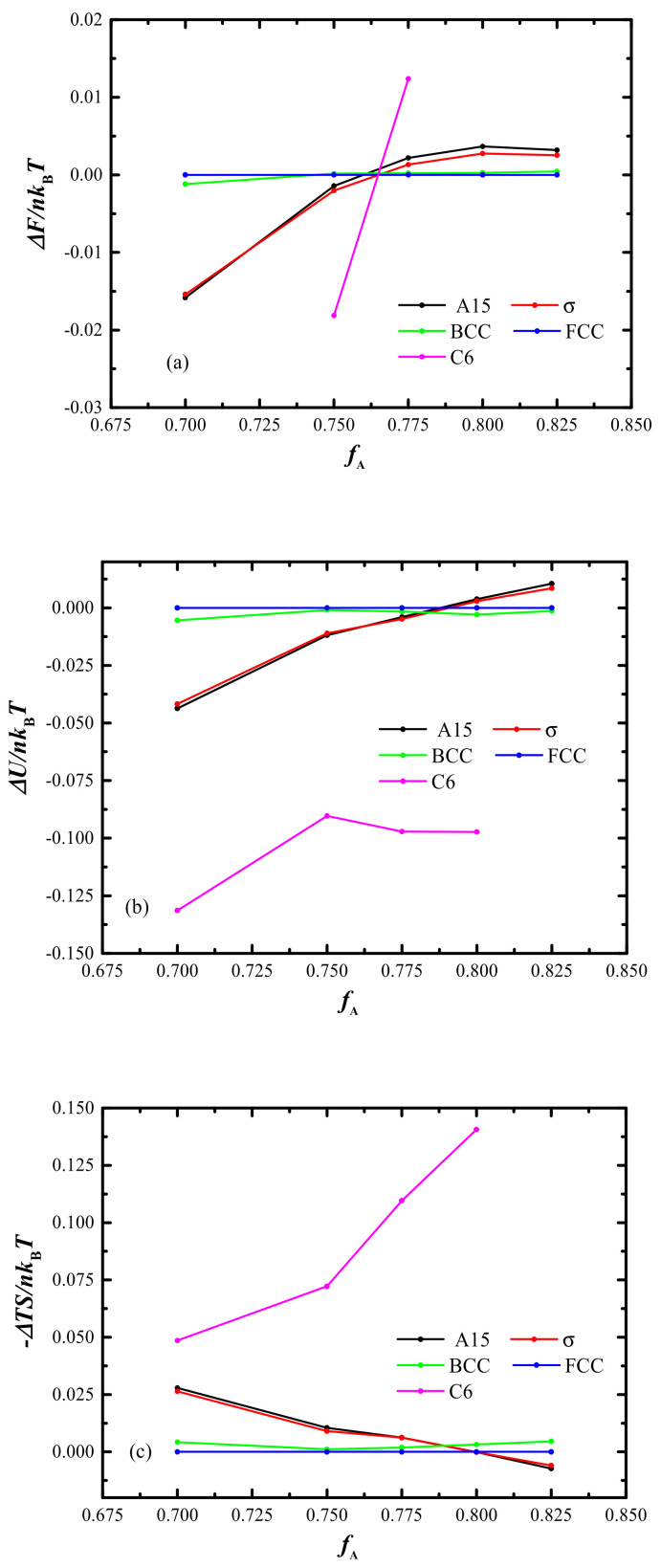
Comparisons of (**a**) free energy, (**b**) interfacial energy, (**c**) entropy energy along changing *f*_A_ of the part of the candidate phases with fixed *τ* = 0.8 and *χN* =80, where the FCC is treated as the reference.

**Figure 9 molecules-28-03536-f009:**
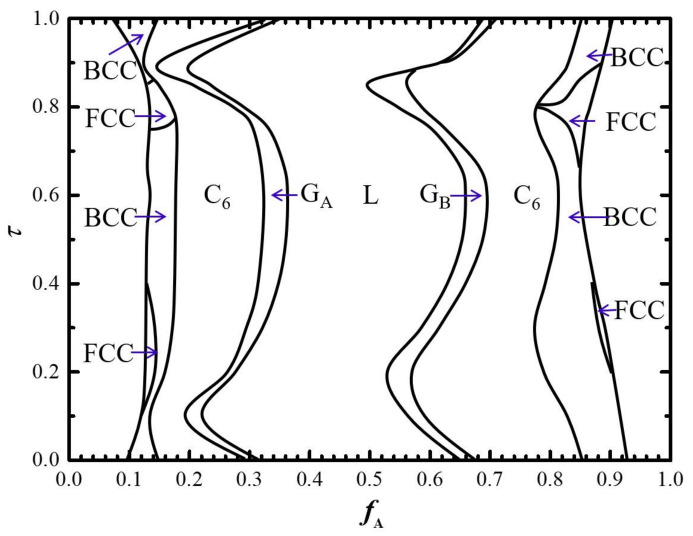
Phase diagram in *f*_A_–*τ* plane for B_1_A_1_B_2_A_2_B_3_ pentablock copolymer with fixed *χN* = 100.

**Figure 10 molecules-28-03536-f010:**
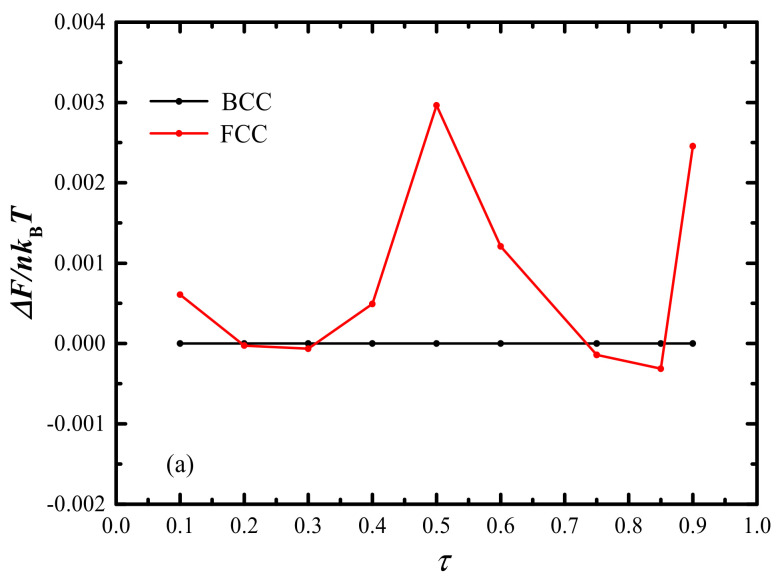
(**a**) Free energy, (**b**) interfacial and entropy energies comparison between BCC and FCC as a function of *τ* (0.1 < *τ* < 0.9) with fixed *χN* = 100 and *f*_A_ = 0.14, where the free energy of BCC is treated as the reference.

**Figure 11 molecules-28-03536-f011:**
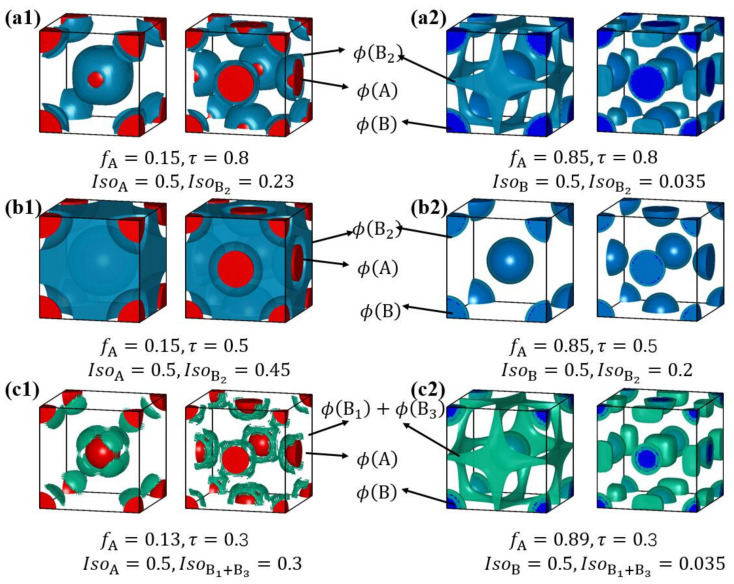
The isosurface plots of different blocks in BCC and FCC phases, where the value of *f*_A_, *τ*, and isovalues (*Iso*) are listed below each subfigure with fixed *χN* = 100 for all the figures. In the left panel, the isosurface of A- and B- blocks are presented for three typical *τ* ( 0.8, 0.5 and 0.3) shown in (**a1**,**b1**,**c1**), where the spherical domain formed from A-component. In the right panel, the isosurface of A- and B- blocks are presented for three typical *τ* ( 0.8, 0.5 and 0.3) shown in (**a2**,**b2**,**c2**), where the spherical domain formed from B-component.

## Data Availability

Not applicable.

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
