# Peer review of "Theoretical Study of Phase Behaviors of Symmetric Linear B1A1B2A2B3 Pentablock Copolymer"

_molecules, 2023, doi:10.3390/molecules28083536_

Round 1

Reviewer 1 Report

The manuscript, "Theoretical study of phase behaviors of symmetric linear Pentablock Copolymers by Zhao et al., describes an SCFT study of phase behavior of a symmetric penta-block copolymer system and the impact of the bridging block on structure formation.  The results indicate a transition from BCC to FCC by tuning the B2 block and thus the importance of consideration of B2 in rational design of supramolecular polymer conformations.

This is a well-developed study with important findings in terms of the impact of B2 length and its important position in driving polymer organization.  I find the study to be imporant, well presented and coherent and worthy of publication.  I have minor edit requests listed below but would support publication of this work.

Minor edits:

1. The English is quite good however I would suggest an editor or reviewer to improve some sentence structure and misplaced words throughout the manuscript.  This is minor but could take a short review.

2. I would remove the conjecture around fig. 11 and Fig 11.  I think this is for a further study and does not add value in its curent state to the overall manuscript.  I would include that in a talk, but not in a publication at this point.

Author Response

1.The English is quite good however I would suggest an editor or reviewer to improve some sentence structure and misplaced words throughout the manuscript.  This is minor but could take a short review.

Reply: Fixed. Thanks for your question. We have carried out a thorough revision on sentence structure and misplaced words throughout the manuscript of the manuscript.   

2.I would remove the conjecture around fig. 11 and Fig 11.  I think this is for a further study and does not add value in its curent state to the overall manuscript.  I would include that in a talk, but not in a publication at this point.

Reply: Fixed. Thanks for your question. The discussion about Figure 11 has been deleted.

Reviewer 2 Report

In this work the author investigated the phase behaviors of a symmetric linear pentablock copolymer by using the self-consistent field theory (SCFT) and the influence of relative length of bridging B2-block on the formation of ordered nanostructures was revealed. Overall, the manuscript is well written and the results sound good. The article is recommended in this journal after addressing minor concerns as listed below.

(1)  The abstract can be written in more interesting way.

(2)  It is proposed to include in the introductory section the work containing the theory and experiment on the phase behavior of BCPs.

Author Response

1. The abstract can be written in more interesting way.

Reply: Fixed. Thanks for your suggestion. We have added some key descriptions of the article in this part and improved the quality of abstract.

2. It is proposed to include in the introductory section the work containing the theory and experiment on the phase behavior of BCPs.

Reply: Fixed. Thanks for your suggestion. We have added some work background containing the theory and experiment on the phase behavior of BCPs